# Toward establishing a rapid constant temperature detection method for canine parvovirus based on endonuclease activities

Shaoting Weng,[1] Shengming Ma,[1] Yueteng Xing,[1] Wenhui Zhang,[1] Yinrong Wu,[1] Mengyao Fu,[2] Zhongyi Luo,[1] Qiuying Li,[1] Sen Lin,[3] Longfei Zhang,[2] Yao Wang[1]

**ABSTRACT** Canine parvovirus (CPV) can cause high morbidity and mortality rates in puppies, posing a significant threat to both pet dogs and the breeding industry. Rapid, accurate, and convenient detection methods are important for the early intervention and treatment of canine parvovirus. In this study, we propose a visual CPV detection system called nucleic acid mismatch enzyme digestion (NMED). This system combines loop-mediated isothermal amplification (LAMP), endonuclease for gene mismatch detection, and colloidal gold lateral chromatography. We demonstrated that NMED can induce the binding of the amplicon from the sample to the specific labeling probe, which in turn triggers digestion by the endonuclease. The sensitivity and visual visibility of LAMP were increased by combining endonuclease and colloidal gold lateral chromatography assisted by a simple temperature-controlled device. The sensitivity of the NMED assay was 1 copy/μL, which was consistent with quantitative PCR (qPCR). The method was validated with 20 clinical samples that potentially had CPV infection; 15 positive samples and 5 negative samples were evaluated; and the detection accuracy was consistent with that of qPCR. As a rapid, accurate, and convenient molecular diagnostic method, NMED has great potential for application in the field of pathogenic microorganism detection.

**IMPORTANCE** The NMED method has been established in the laboratory and used for CPV detection. The method has several advantages, including simple sampling, high sensitivity, intuitive results, and no requirement for expensive equipment. The establishment of this method has commercial potential and offers a novel approach and concept for the future development of clinical detection of pathogenic microorganisms.

**KEYWORDS** canine parvovirus, endonuclease, loop-mediated isothermal amplification, specific labeling probe, lateral chromatography

Canine parvovirus (CPV) is a single-stranded linear DNA virus belonging to the *Parvoviridae* family, the *Protoparvovirus* genus. This virus is transmitted through direct or indirect contact, which can cause severe infectious intestinal diseases and myocarditis in dogs. This leads to high morbidity and mortality rates in puppies (1, 2). This year, there has been a high incidence trend in China. CPV-2A, CPV-2B, CPV-2C, and their variants have been reported throughout the country (3, 4). The disease not only affects the lives and health of domestic pet dogs but also causes significant losses to the dog industry and wildlife protection industry.

At present, the commonly used methods for detecting canine parvovirus in pet shops and farms are immune colloidal gold strip detection combined with routine blood tests or polymerase chain reaction (PCR) technology (5, 6). However, the immunocolloidal gold assay is not sensitive enough, and there is cross-reaction of antigens. PCR detection has high sensitivity and strong specificity, but it still has some drawbacks, including false positives, high costs, and the requirement for skilled personnel. These

Address correspondence to Longfei Zhang, 879021836@qq.com, or Yao Wang, 695620016@qq.com.

The authors declare no conflict of interest.

See the funding table on p. 12.

factors make it difficult to meet the current clinical needs. Therefore, there is an urgent need for a convenient, rapid, and efficient method of CPV detection that can facilitate early intervention and treatment and reduce the risk of disease transmission. To address the aforementioned issues, many detection methods based on isothermal nucleic acid amplification technology have been developed for the detection of viruses. Cho et al. utilized primers that targeted the VP2 gene of CPV to develop a loop-mediated isothermal amplification (LAMP) detection method as a potential alternative to quantitative PCR (qPCR) (7). Geng et al. also developed a real-time recombinase polymerase amplification (RPA) method for the detection of CPV-2 using primers and specific probes targeting the CPV-2 capsid protein gene (8). However, the single method is highly affected by reaction conditions and has low specificity.

In recent years, isothermal amplification technology has been applied in combination with the CRISPR/Cas system in molecular diagnosis and therapy. Janice et al. developed the DNA endonuclease-targeted CRISPR *trans* reporter detection technology. By reverse-transcribing the sample RNA into DNA and then amplifying it at a constant temperature, the genetic sequence of the virus can be detected. This sequence can then be targeted and cleaved by CRISPR/Cas12 to confirm the presence of the virus (9). In addition, Zhang Feng's team developed the Specific High Sensitivity Enzymatic Reporter UnLOCKing detection technology, which utilizes reverse transcription (RT), RT-LAMP, CRISPR/Cas13 technology coupling, and a magnetic bead enrichment method to optimize nucleic acid extraction. Finally, through test strip detection, they were able to achieve one-step nucleic acid detection (10). Although the detection method based on nucleic acid isothermal amplification combined with CRISPR/Cas has rapidly developed with advantages such as speed, sensitivity, and the absence of complex laboratory equipment, the Cas detection method still cannot be effectively promoted due to its biological reaction characteristics, such as the high design requirements of sgRNA and non-specific enzyme digestion (11, 12).

In this study, a highly sensitive visual detection method of canine parvovirus based on endonuclease [nucleic acid mismatch enzyme digestion (NMED)] was established. CPV DNA extracted from the sample to be tested was amplified using LAMP. It was then hybridized with a specific labeling probe (FAM-Bio) containing a two-base mutation. Finally, specific endonuclease [such as T7 endonuclease I (T7E I)] was added to the mixture. Once the endonuclease recognizes the hybridization sequence between the sample DNA and the DNA probe, it cleaves the labeling probe at the mutation site and subsequently generates a visually distinguishable alteration on the colloidal gold lateral chromatography paper (Fig. 1). This nucleic acid detection method boasts advantages such as convenience, high sensitivity, and intuitive results. Compared to traditional methods, it does not require expensive instruments or complex target primers, effectively reducing detection costs. More importantly, the establishment of this method provides a brand new approach and development direction for pathogen detection, holding significant commercial potential.

## MATERIALS AND METHODS

### Materials

The StarSpin Fast Virus DNA/RNA Kit was purchased from Beijing KangrunChengye Biotechnology Co., Ltd. The pcDNA3.1 plasmid was purchased from Chengdu Chuanshi Biotechnology Co., Ltd. The dNTP Mix and Bst II DNA Polymerase (Bst II DNA Pol) Large Fragment were purchased from Nanjing Nuoweizan Biotechnology Co., Ltd. Betaine was purchased from Beijing Soleibao Biotechnology Co., Ltd. The 2× PCR mixture was purchased from Sangon (Shanghai) Co., Ltd. T7 endonuclease I was purchased from Beyotime Biotechnology Co., Ltd. Genloci Cruiser mutation test kit was purchased from Jiangsu Jirui Biotechnology Co., Ltd. The 2× Taq Master Mix and ChamQ Universal SYBR qPCR Master Mix were purchased from Nanjing Vazyme Biotechnology Co., Ltd. The nucleic acid products of colloidal gold detection cards were custom purchased from

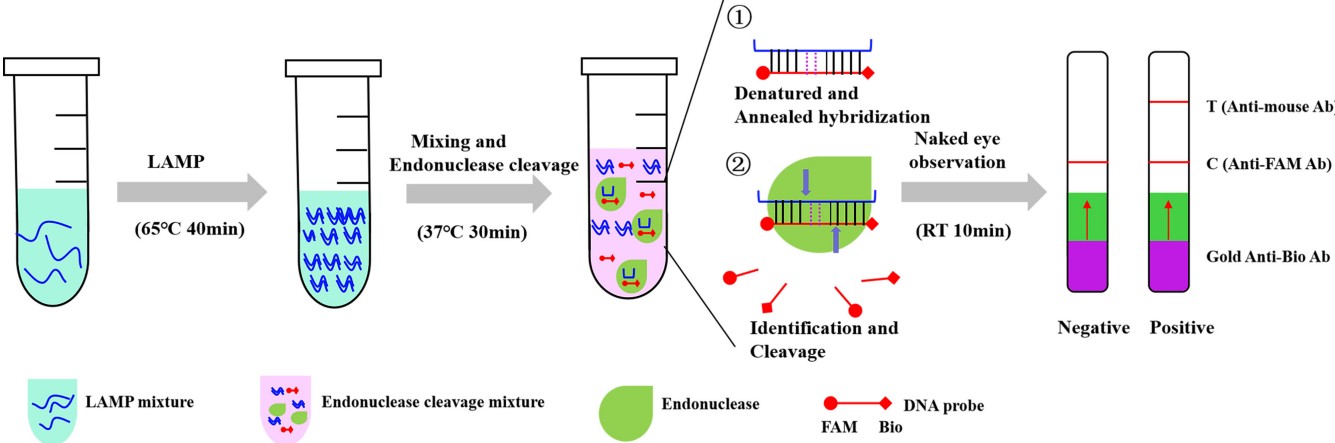

**FIG 1** The scheme of visual nucleic acid mismatch enzyme digestion detection method. For CPV detection, DNA templates extracted from the stool sample are added into the LAMP mixture. After LAMP amplification, the reaction products are mixed with specific labeling probes (including two-base mutation), ① denotes denatured and annealed hybridization. ② denotes that once endonuclease recognized the annealed product, it splits the prodcut directly, generating a signal visible to the naked eye under colloidal gold lateral chromatography paper.

Nanjing Zhongding Biotechnology Co., Ltd. The CPV was provided by pet clinics in Anyang City and surrounding areas. It was taken from anal test strips and stool samples of affected dogs and suspected cases.

## Design and synthesis of probes and primers

By comparing and analyzing the conserved sequence of the canine parvovirus VP2 gene published by GenBank (GenBank: OR667804.1), we designed and examined several primer sets for LAMP and qPCR analysis and the corresponding double-labeling DNA probe. These included the CPV2 full-length primer CPV-CDS, as well as four LAMP primer sets targeting different sites: CPV1, CPV2, CPV3, and CPV4. Additionally, we used the qPCR identification primer CPV-qPCR. The same template DNA and other LAMP reactants were mixed with four LAMP primers and placed in a constant temperature instrument for sequence amplification in order to select the best primers in different combinations. Two double-labeling DNA probes, CPV2-TZ and CPV3-TZ, were synthesized based on the specific site of the target sequence. The middle site of the 30-bp sequence was designed with a two-base mutation (underlined). The nucleic acid sequences of primers and probes are shown in Table 1. They were synthesized by Sangon (Shanghai) Co., Ltd.

## DNA extraction for CPV

The extraction method was carried out following the operating instructions of the StarSpin Fast Virus DNA/RNA Kit (P144-01; Beijing Kangrunchengye Biotechnology Co., Ltd.). The viral DNA was collected and stored at −20°C. The gene sequence was amplified using 2 µL of CPV DNA (5,319 bp) and CPV-CDS primer, following the operating instructions of 2× Taq Master Mix (P111-01; Nanjing Vazyme Biotechnology Co., Ltd.). The amplified PCR product was then ligated to the pcDNA3.1 vector. After that, it was transformed into the DH5a competent cell and sequenced for identification, resulting in the production of the pcDNA3.1-VP2 plasmid. The original DNA concentration of VP2 was determined using the NanoDrop ND-1000 (Thermo Fisher Scientific Inc., Waltham, MA, USA) and diluted to $1 \times 10^7$ copies/µL.

## qPCR assay

ChamQ Universal SYBR qPCR Master Mix was used for qPCR detection of the DNA being tested. The reaction system of qPCR was 20 µL, consisting of 10-µL 2× SYBR Mix, 1-µL (10-µM) primers CPV-qPCR-F and CPV-qPCR-B, 1-µL pcDNA3.1-VP2 plasmid/CPV

**TABLE 1** Primer and probe nucleotide sequences

| Name | Sequences (5′–3′) |
| --- | --- |
| CPV-CDS-F | ATG AGT GAT GGA GGA GTT CA |
| CPV-CDS-R | TTT CTA GGT GCT AGT TGA GA |
| **CPV1-F3**[a] | GTA AAC CAT GTA GAC TAA CAC AT |
| **CPV1-B3**[a] | GCA CTA TAA CCA ACC TCA GC |
| **CPV1-FIP**[a] | GCT CCT TCA GAT TGA GGC AAA GAC ATG GCA AAC AAA TAG AGC A |
| **CPV1-BIP**[a] | GAG TTC AAC AAG ATA AAA GAC GTG GGG TCT CAT AAT AGT AGC TTC AGT |
| **CPV1-LB**[a] | ATT TAG AAA TGG TGG TAA GCC CAA |
| **CPV2-F3**[a] | TGC CTC AAG CTG AAG GAG |
| **CPV2-B3**[a] | TTT CAT ATG TTT GCG CTC C |
| **CPV2-FIP**[a] | TCC CAT TTG AGT TAC ACC ACG TCT ACT AAC TTT GGT GAT ATA GGA G |
| **CPV2-BIP**[a] | TTA TGA GAC CAG CTG AGG TTG GGT GTT TTA AAT GGC CCT TGT G |
| **CPV2-LB**[a] | GCA CCA TAT TAT TCT TTT GAG GCG T |
| **CPV3-F3**[a] | TGG GAT AAA GAA TTT GAT ACT GA |
| **CPV3-B3**[a] | TGA GAG GCT CTT AGT TTA GCT TT |
| **CPV3-FIP**[a] | CGC AAC TTT TAC AAA TAA TTG ACC ATT AAA ACC AAG ACT TCA TGT AAA TGC |
| **CPV3-BIP**[a] | AAC AAA TGA ATA TGA TCC TGA TGC AAC CTT TCC ACC AAA AAT CTG AGT |
| **CPV3-FLP**[a] | GGA CAA TTA TTT TGA CAA ACA AAT GG |
| **CPV3-BLP**[a] | TCT GCT AAT ATG TCA AGA ATT GTA AC |
| **CPV4-F3**[a] | GGT ACA GAT CCA GAT GAT GTT C |
| **CPV4-B3**[a] | GCA CTA TAA CCA ACC TCA GC |
| **CPV4-FIP**[a] | GCC ATG TAT GTG TTA GTC TAC AAT TCT GTG CCA GTA CAC TTA C |
| **CPV4-BIP**[a] | GCA TTG GGC TTA CCA CCA TTT CCT GTT GAC TCC TAT ATC ACC |
| **CPV4-FLP**[a] | TCC TGT AGC AAA TTC ATC ACC T |
| **CPV4-BLP**[a] | TGC CTC AAT CTG AAG GAG C |
| **CPV2-TZ**[b] | GAC GTG GTG TA<u>A C</u>AT CAA TG |
| **CPV3-TZ**[b] | TGA TCC TGA CG<u>C A</u>TC TGC TA |
| **CPV-qPCR-F**[c] | GAA TCT GCT ACT CAG CCA CCA AC |
| **CPV-qPCR-R**[c] | GTG CAC TAT AAC CAA CCT CAG C |

[a]CPV 1~4 -F3、CPV 1~4 -B3、CPV 1~4 -FIP、CPV 1~4 -BIP、CPV 1~4 -FLP/LB、CPV 1~4 -BLP/LB: CPV means canine parvovirus, 1~4 represents the first~fourth set of primers designed on the VP2 gene, F3, B3, FIP, BIP, FLP, BLP (LB) represent F3, B3, FIP, BIP, LoopF, LoopB primers in LAMP reaction system, respectively.
[b]CPV 2/3 -TZ:represents the probe corresponding to the second/third set of LAMP primers of canine parvovirus.
[c]CPV-qPCR-F/R:represents the Forward/Reverse primer for quantitative fluorescent PCR of canine parvovirus.

DNA, and 7-µL RNase-free $H_2O$. Nucleic acid was quantitatively detected using QIAGEN PCR kit. The reaction procedure involved predenaturation at 95°C for 30 seconds. The amplification process consisted of 40 cycles, with each cycle consisting of 95°C for 10 seconds and 60°C for 30 seconds. After amplification, the data were copied and analyzed. Subsequently, 10 µL of the amplified product was subjected to 1% agarose gel electrophoresis at 120 V for 30 min.

## LAMP and endonuclease digestion conditions

Sample DNA was collected using the viral DNA extraction method described in DNA Extraction for CPV. The LAMP reaction consisted of 5 µL of 10× IsothermalAmp Buffer (10-mM Tris-HCl, pH 7.5, 50-mM KCl, 0.1-mM EDTA, 0.1% TritonX-100, and 50% glycerin), 3 µL of (100 mM) $MgSO_4$, 7 µL of (10 mM) dNTP mix, 2-µL of (8-U/µL) Bst II DNA Pol, 11.5 µL of RNase-free $H_2O$, 1 µL of pcDNA3.1-VP2 plasmid or 2 µL of sample DNA, 5 µL of (10 mol/L) betaine, and 2 µL of (40 µM) CPV-FIP and CPV-BIP primers, 3 µL (5 µM) of CPV-F3 and CPV-B3 primers, followed by 2 µL (20 µM) of CPV-FLP and CPV-BLP primers, or 2 µL (40 µM) of CPV-LB primers were added. The reaction tube, with all samples added, was mixed in the oscillating vortex and then promptly centrifuged. The reaction mixture was incubated at 45°C for 40 min. After amplification, 10 µL of the amplified product was loaded onto a 1% agarose gel and subjected to electrophoresis at 120 V for 30 min.

The LAMP reaction products were taken in 2 µL and mixed with the appropriate proportion of 10× reaction buffer (500-mM NaCl, 100-mM Tris-HCl, 100-mM MgCl$_2$, 10-mM DL-Dithiothreitol (DTT), pH 7.9) and CPV-TZ probes (including two-base mutation). Then, they were denatured, annealed, and renatured (98℃ for 3 min, 98℃–85℃ for 2℃/s, 85℃–25℃ for 0.1℃/s), respectively. Then 2 U of T7E I was added, and the reaction was carried out at 37℃ for 20 min. T7E I was able to identify sites where probes have been mismatched with LAMP products. Ten microliters of the reaction products was analyzed in 1% agarose gel electrophoresis. The remaining products were detected using colloidal gold lateral chromatography paper.

## Optimization of reaction conditions

NMED can be successfully performed over a wide range of reaction temperatures and incubation times. Scale-up evaluation of efficiency under various conditions can enhance the performance of NMED analysis. Therefore, the LAMP tests in this experiment were performed at different incubation temperatures (65℃, 60℃, 55℃, 50℃, 45℃, 40℃, and 35℃) and incubation times (10, 20, 30, 40, 50, and 60 min). The LAMP products were detected using T7E I and Cruiser enzymes, following the optimized procedure mentioned above. The optimal reaction temperature, time, and endonuclease enzyme were selected by analyzing the CPV reaction products using 1% agarose gel electrophoresis.

## Sensitivity evaluation

DNA of pcDNA3.1-VP2, with a concentration of $1 \times 10^7$ copies/µL, was diluted by a 10-fold ratio with distilled water. Eight gradient concentrations (from $10^7$ to 1）were used. Then, LAMP tests were conducted using the optimal reaction conditions outlined in Optimization of Reaction Conditions, along with the qPCR test. Subsequently, 10 µL of the resulting products was analyzed using 1% agarose gel to determine the detection limit of the LAMP method. The residual products were digested by the T7E I, and the detection limit was confirmed using lateral chromatographic test paper.

## Testing principle

The NMED enhanced T7E I immunochromatographic detection system utilizes the FAM-biotin reporter gene to facilitate on-site visual detection. In the negative samples, the gold anti-biotin antibody was fully bound to the high concentration FAM-biotin reporter gene, and the conjugate was intercepted by the anti-FAM antibody in the control band (control line, C). For positive samples, the FAM-biotin reporter genes were cleaved, and gold grains were conjugated with anti-biotin antibodies accumulated in the detection band (detection line, T) while decreasing in the control band. Gray-scale analysis of stained bands using Photoshop software allowed us to obtain a more precise understanding of band intensity. The schematic diagram of immunochromatographic detection is shown in Fig. 1.

## Detection NMED performance in clinical samples

The applicability of NMED testing for CPV was verified by using clinical samples from 20 confirmed or suspected cases collected at Anyang Pet Hospital in Henan Province from June to August 2023. DNA was extracted from the samples using the viral DNA extraction method described in DNA Extraction for CPV. After the LAMP reaction, 2 µL of nucleic acid was used as the template and the probe was added. The hybrid products, after denatured annealing, were enzymatically digested with an endonuclease. The colloidal gold lateral chromatography paper tests were performed on the digested products, and the results were interpreted after a 10-min reaction. Clinical samples were also evaluated by a qPCR assay. The results of NMED were compared with those of qPCR.

## RESULTS

### Feasibility of NMED

We initially tested four pairs of LAMP primers that targeted various conserved domains of the VP2 gene. The findings revealed that the CPV-2 and CPV-3 primers produced more amplified products compared to the other two pairs of LAMP primers in the agarose gel (Fig. 2A). Therefore, in the following experiments, we selected CPV-2 and CPV-3 primers to verify the feasibility of NMED. We combined the LAMP products with the specific probe CPV2-TZ or CPV3-TZ, which corresponded to the CPV sequence. We then mixed them with T7E I in a centrifuge tube and analyzed the digested product on a PAGE gel.

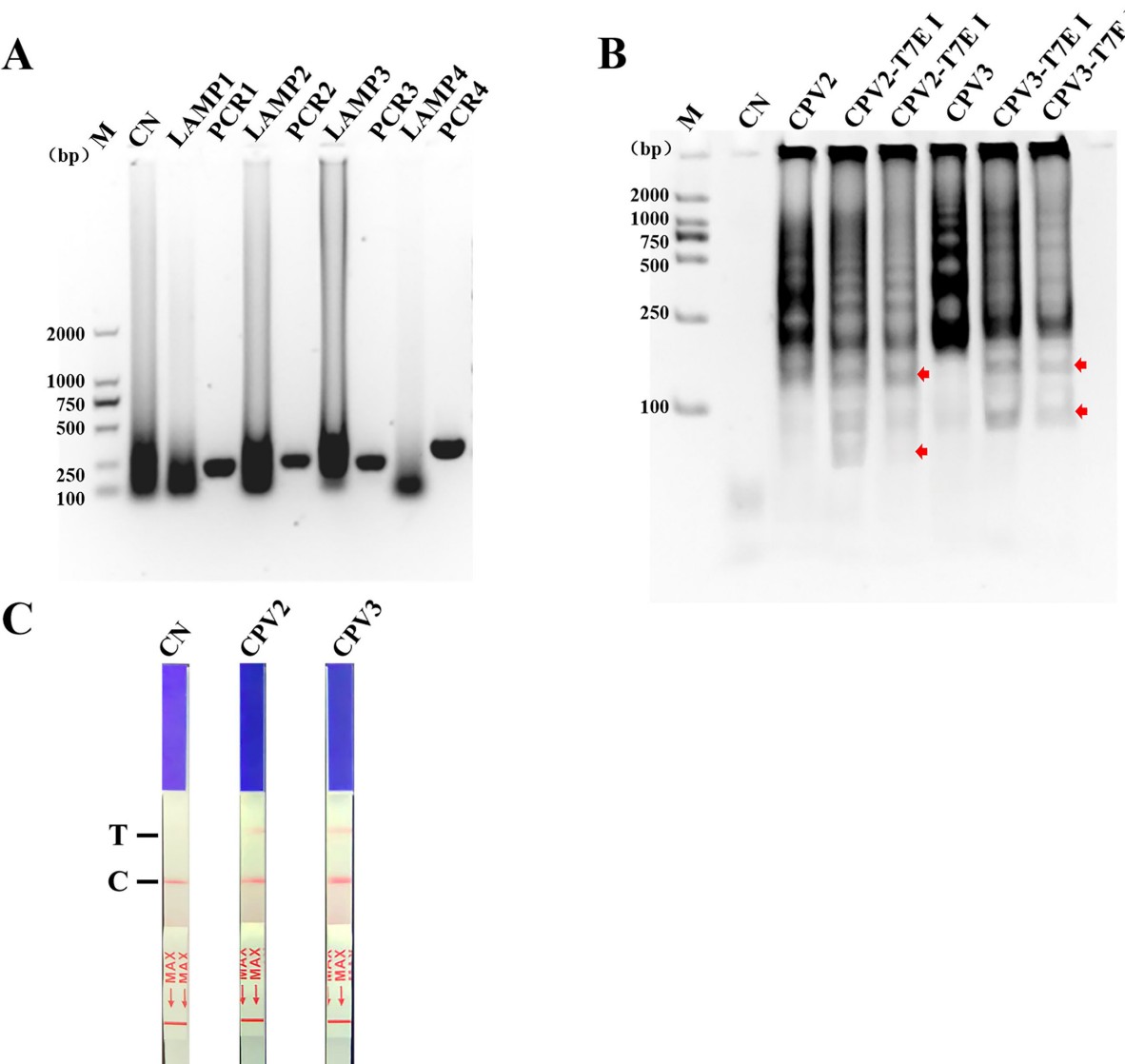

FIG 2  Results of NMED feasibility analysis. (A) CPV was detected using LAMP with different primers. LAMP1–4 are the amplification products of CPV-1, CPV-2, CPV-3, and CPV-4 primers for LAMP; PCR1–4 are the result of F3/B3 amplification products of CPV-1, CPV-2, CPV-3, and CPV-4 primers for PCR; CN is positive sample result used as a control for LAMP. (B) T7E I enzymatic identification of LAMP products of CPV. CPV2 and CPV3: the amplification products of CPV-2 and CPV-3 amplification products were not cleaved by T7E I; CPV2-T7E I: the CPV-2 amplification products were cleaved by T7E I; the enzyme cleaved fragments are indicated by red arrows; CPV3-T7E I: the CPV-3 amplification products were cleaved by T7E I; the enzyme cleaved fragments are indicated by red arrows. (C) Lateral chromatographic paper test for detecting enzyme cleaved products. CPV2: the CPV-2 cleaved products were detected by lateral chromatographic paper test; CPV3: the CPV-3 cleaved products were detected by lateral chromatographic paper test. C, control line; CN, distilled water was used for adding controls; M, 2,000-bp marker; T, detection line.

The results show a significant change in the band size of T7 enzyme cleavage products compared to the control group and the appearance of noticeable smaller bands. This indicates that T7E I was able to recognize and cleave CPV-2 and CPV-3 products (Fig. 2B). The results of the lateral chromatographic test paper showed that the CPV-2 and CPV-3 cleavage products could be detected on the detection line. The band density analysis revealed that the detection of CPV-3 was superior to that of CPV-2 (Fig. 2C). It was indicated that the double-labeled probes were detected on the lateral chromatographic test paper after cleavage by T7E I enzyme. This allows us to conclude that the cleavage products contains the target gene. The samples to be tested could be identified using LAMP, endonuclease identification and cleavage, and lateral chromatographic test paper, which proved the feasibility of the NMED method.

## Optimization of the LAMP method for CPV detection

We amplified the plasmid standard using LAMP, mixed it with Bst enzyme, and incubated it at a temperature range of 65℃–35℃ for 60 min. The results of agarose gel electrophoresis showed that the target product band was evident from 65℃ to 50℃, but its brightness gradually increased. This could be due to the increase in enzyme activity with decreasing temperature. At 45℃, the band was clearly visible. At 40℃, the bands were faint and unclear. No bands were observed at 35℃ (Fig. 3A). In the subsequent time-course experiment, the results were detected every 10 min after incubation at 45℃ for different durations, ranging from 0 to 60 min. The results of agarose gel electrophoresis showed that there were no bands at 0, 10, and 20 min. From 30 to 60 min, the bands changed from light and fuzzy to dark and clear. The reaction results were better at 40, 50, and 60 min (Fig. 3B). Therefore, in LAMP amplification, the optimal temperature for the Bst enzyme was 45℃, and the reaction time was 40 min.

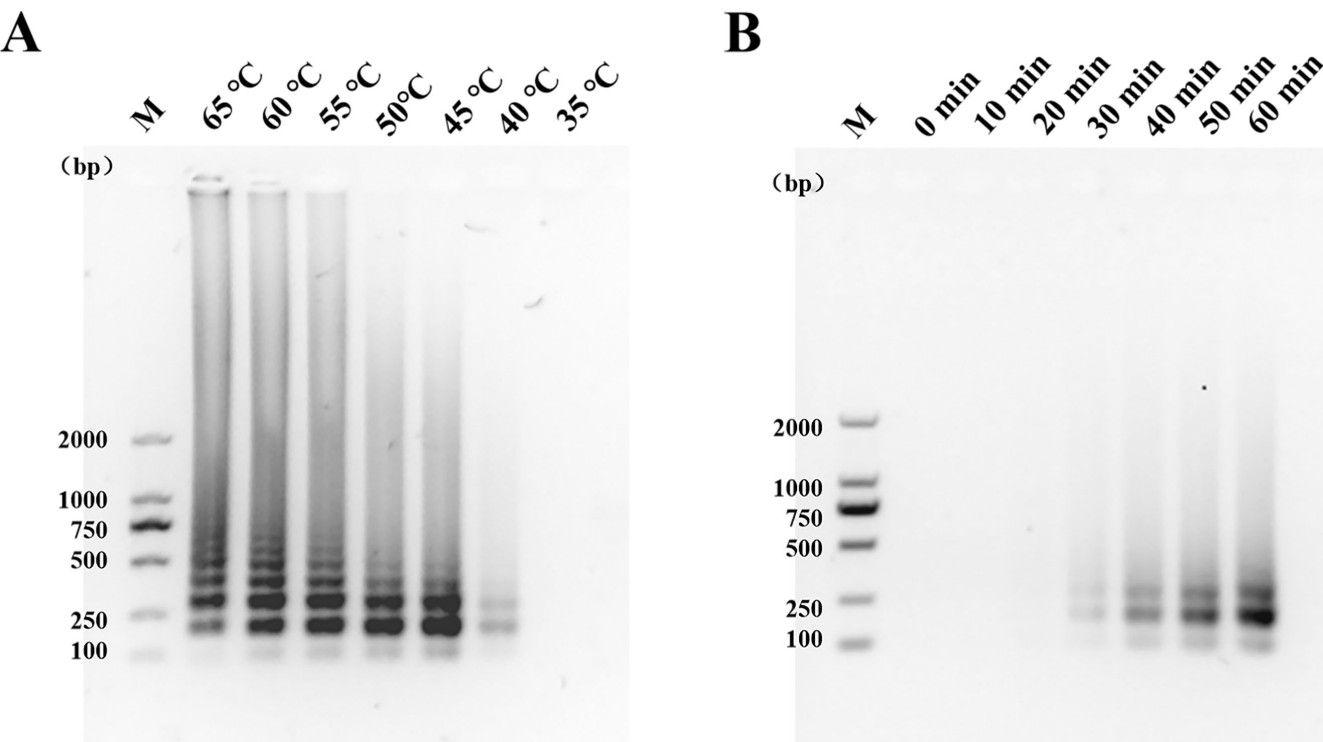

**FIG 3** Optimization of the LAMP assay. (A) Various incubation temperatures were evaluated. The temperatures were 65℃, 60℃, 55℃, 50℃, 45℃, 40℃, and 35℃. (B) Various incubation times were evaluated. The time intervals were 0, 10, 20, 30, 40, 50, and 60 min. M, 2,000-bp marker.

## Optimization of enzyme cleavage and lateral chromatography

The LAMP amplified products were denatured and then annealed with specific labeling probes. The samples were then combined with T7E I/Cruiser and incubated at 37°C/45°C for 20 min, respectively. The results of agarose electrophoresis showed that the positions of the product bands shifted after enzyme cleavage. Both T7E I and Cruiser were able to cleave the hybrid of the amplified product and the specific labeling probes, and Cruiser exhibited a more effective cleavage compared to T7E I (Fig. 4A). A colloidal gold lateral chromatography paper was used to detect the cleaved products. The results indicated that the cleaved labeling probes could be identified in the reaction products. The cracking products were detected by colloidal gold lateral chromatography. The band density analysis showed that Cruiser was superior to T7E I in cleave efficiency (Fig. 4B).

## Sensitivity assessment of NMED assay

Next, we investigated the sensitivity of NMED by testing eight plasmid standards using a 10-fold serial dilution ratio. We compared the results of qPCR with those of NMED and found that the sensitivity of NMED was similar to that of qPCR (Fig. 5A and B). Agarose gel analysis of the amplified product revealed that the detection limit of LAMP, visible to the naked eye, was 10 copies (Fig. 5C). After combining the LAMP product with the specific labeling probe, the detection limit of NMED was determined to be one copy through endonuclease cleavage and gold lateral chromatography paper (Fig. 5D). The results showed that NMED improves the sensitivity of LAMP, thereby enhancing the detection capability to the level of a single molecule.

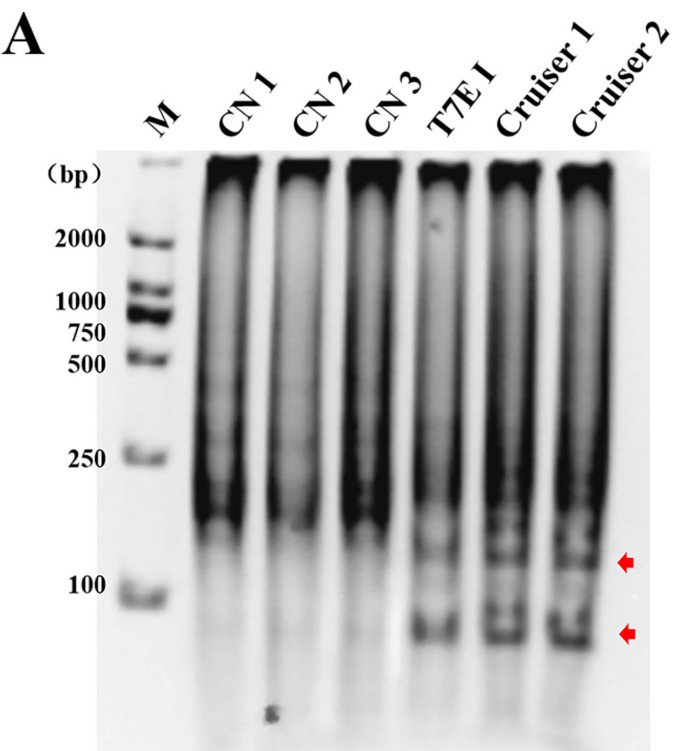
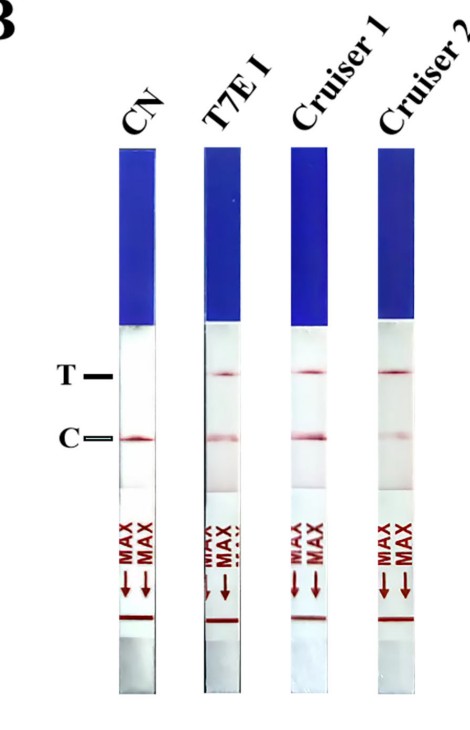

**FIG 4** Identification of enzyme cleavage results by LAMP method. (A) Identification of endonuclease cleavage in LAMP products of CPV. CN1-3: LAMP products of CPV-3 were not cleaved; T7E I: LAMP products of CPV-3 were cleaved by T7E I; Cruiser 1 and Cruiser 2: LAMP products of CPV-3 were cleaved by Cruiser; the enzyme cleaved fragments are indicated by red arrows. (B) Lateral chromatographic paper test for the detection of endonuclease cleavage products. CN: LAMP products of CPV-3 were not cleaved; T7E I: LAMP products of CPV-3 were cleaved by T7E I; Cruiser 1 and 2: AMP products of CPV-3 were cleaved by Cruiser. C, control line; M, 2,000-bp marker; T, detection line.

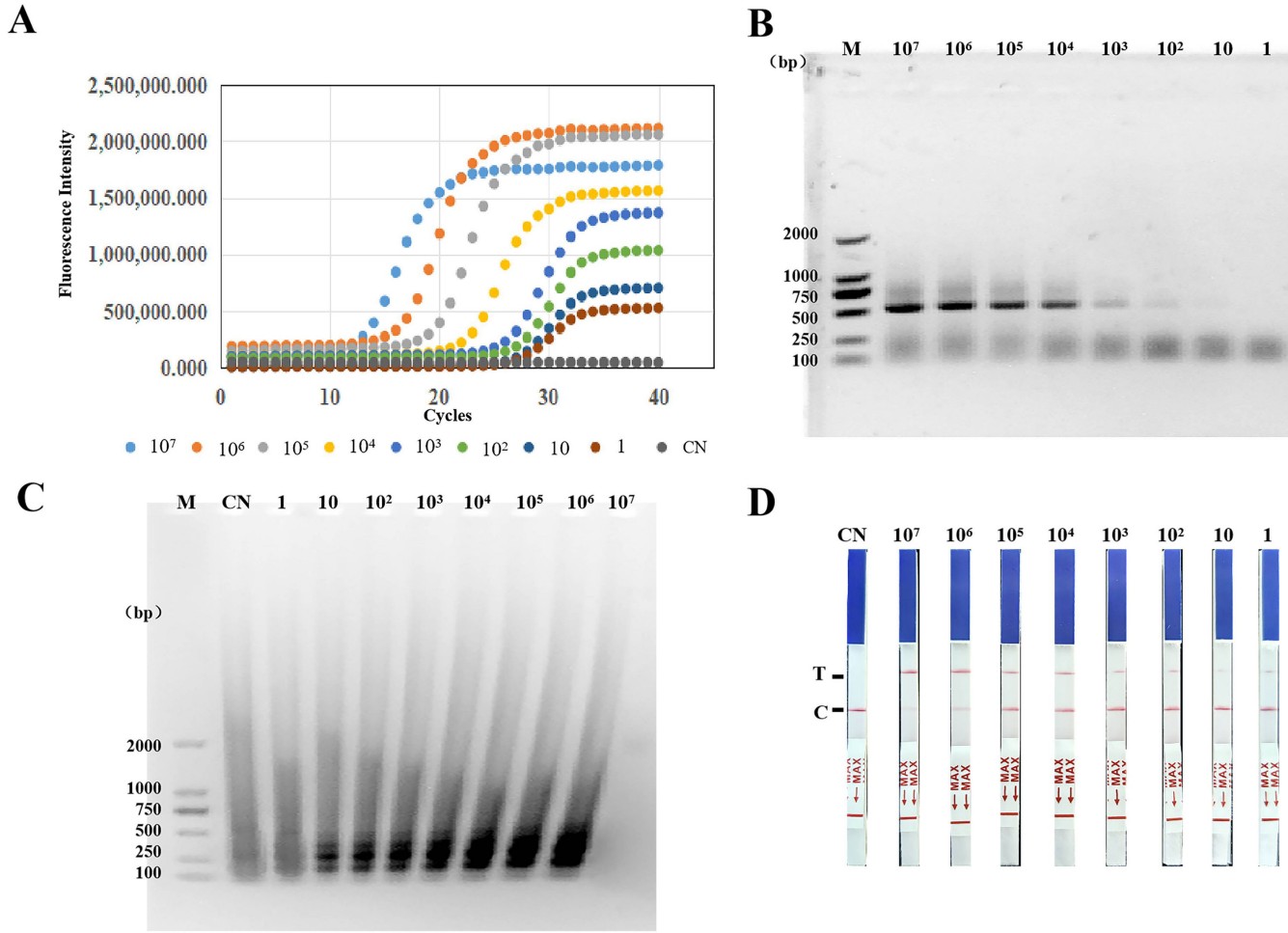

**FIG 5** Sensitivity detection results of NMED and qPCR.(A) Expression results of fluorescence intensity from qPCR. (B) Results of agarose gel electrophoresis of qPCR-amplified products. (C) Results of agarose gel electrophoresis of LAMP amplified products. (D) Results of lateral chromatographic paper test for NMED cleavage products. Ten-fold serial dilutions of the DNA standard were prepared, ranging from $10^7$ to 1 copy/µL. CN: distilled water was used for adding controls; M, 2,000-bp marker; T, detection line, C, control line.

## The accuracy and reliability of NMED assay

In order to assess the accuracy and reliability of NMED detection of CPV, fecal and anal swab samples from 20 suspected cases of clinical infection were tested using this method. qPCR was used as the control method. The DNA extracted from the samples was used as a template for NMED and qPCR detection. The results showed that 15 samples tested positive for CPV using NMED and qPCR, while 5 samples tested negative using NMED and qPCR (Fig. 6). The results showed that NMED and qPCR were consistent with each other, and the coincidence rate was 100%, indicating the method could be used for CPV detection.

## DISCUSSION

Nowadays, CPV and its variants have been circulated in carnivores and have been isolated and identified in several countries worldwide. It has a strong infectivity, inhibits the immune system, causes serious digestive system and myocardial diseases, and continues to cause significant harm worldwide. The genetic mutation of the CPV gene also poses significant challenges for clinical identification and disease prevention and control. Therefore, the demand for rapid and efficient diagnosis of CPV is increasing (13, 14).

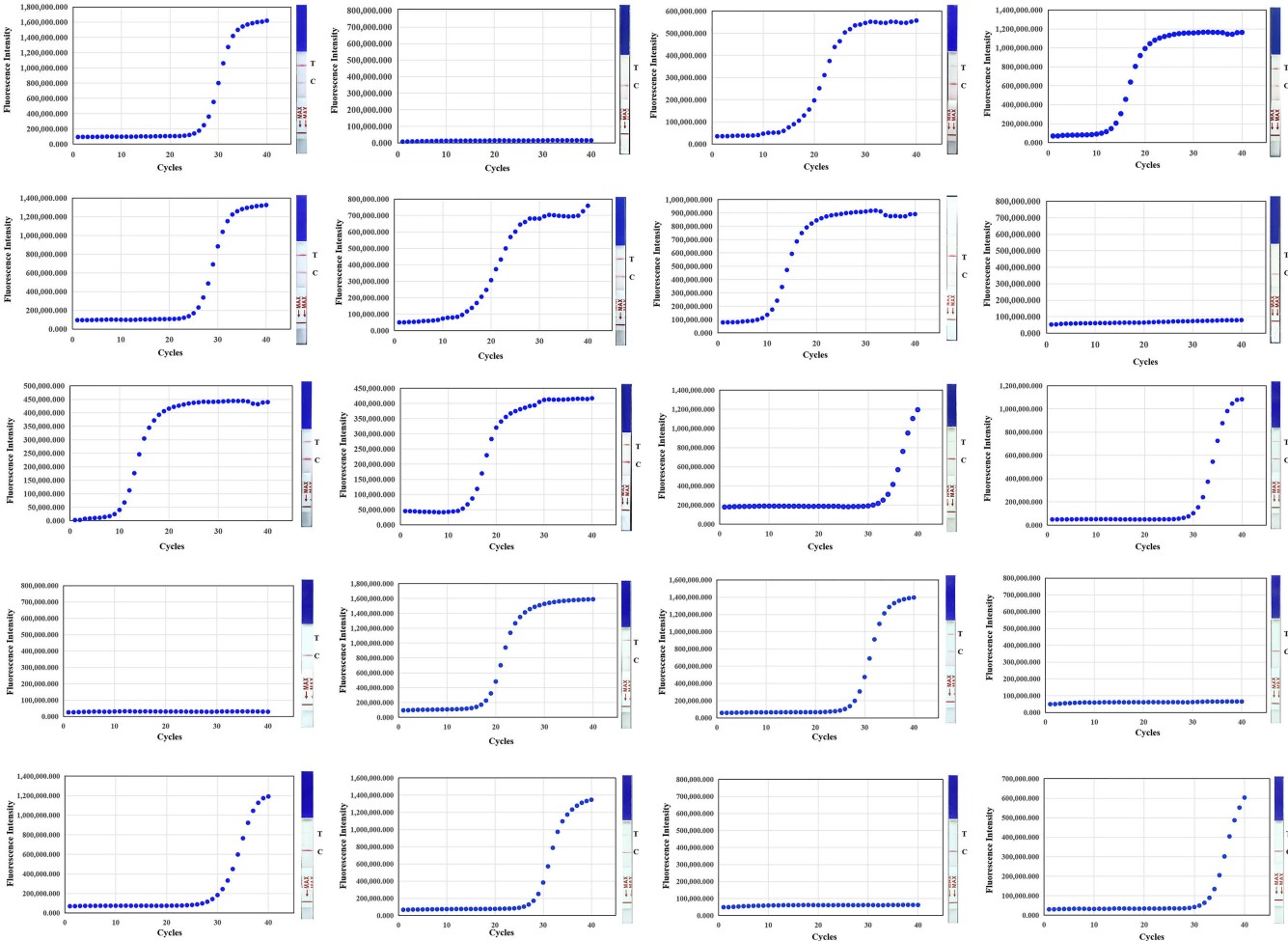

**FIG 6** CPV detection in 20 suspected cases with NMED and qPCR. Results of qPCR and NMED were used for each suspected case. C, control line; T, detection line.

Isothermal amplification technology still has obvious advantages in various fields, including medical health and animal disease pathogen detection, among others, and it is constantly improving (15, 16). Rapid detection based on isothermal amplification of nucleic acid usually involves three main steps: nucleic acid extraction, isothermal amplification, and product detection (17). Each step requires solving various problems, such as simplifying and improving the efficiency of nucleic acid extraction and amplification, eliminating constraints imposed by instruments and limited field environments, expanding the reaction temperature range of isothermal amplification, enhancing the stability of the reaction system, and improving the sensitivity and visualization of product detection. In addition, the technology can also leverage its own advantages by combining microfluidic chips, nano-gold, CRISPR, and other technologies (18–20) to establish more rapid, accurate, and simple detection methods. This will provide better services for animal disease detection.

Many researchers have optimized nucleic acid amplification techniques to make them more user-friendly. One-step detection technology based on RPA and CRISPR/Cas is becoming increasingly mature. A developed rapid and visual detection one-step RPA-CRISPR detection (ORCD) system for nucleic acids enables simultaneous Cas12a detection and RPA amplification without separate preamplification and product transfer steps. It can detect 0.2 copy/µL of DNA and 0.4 copy/µL of RNA. The system can extract, amplify, and detect samples within 30 min, with a sensitivity and specificity of more than 97.30% (21). However, the presence of various enzymes in the RPA system greatly affects

its reaction conditions, leading to a high background in visual detection due to the short reaction time (22). In addition, the amplicon yield of RPA is lower than that of LAMP, which in turn reduces the sensitivity of CRISPR detection. The one-pot visual severe acute respiratory syndrome coronavirus 2 (SARS-CoV-2) detection system, "opvCRISPR," was developed by Wang et al. It integrates LAMP and Cas12a in a reaction system to complete the detection. The incubation time after combining the two methods was 120 min, and the process was carried out in two separate reactions. The reaction results were obtained with a minimum concentration of five copies (23). A colorimetric detection method Cas12a-assisted RT-LAMP/AuNP (CLAP) based on Cas12a-assisted LAMP/AuNP was developed for rapid and sensitive detection of SARS-CoV-2. Under optimal conditions, 4 copies/µL of SARS-CoV-2 RNA could be detected by the naked eye within 40 min (24).

The detection based on the CRISPR/Cas12 system has several advantages, including signal amplification, high sensitivity, and high efficiency. However, it also has some limitations, such as the high design requirements for primers, non-specific Cas12 lateral enzyme digestion, and the interference of off-target effects in sequence-based detection (25, 26). Therefore, we propose a novel method called NMED, which combines LAMP amplification, specific probe hybridization, and endonuclease-specific cleavage. In this NMED method, the required selection area of primers is large; the design is simple; and the application is easy. Our proposed method utilizes the endonuclease's characteristic to identify the mutation site. It significantly enhances detection specificity by identifying the mutant base in the hybridized sequence of the specific targeted sample DNA and specific labeling antibody. This method is more sensitive.

However, NMED, like other nucleic acid detection techniques, is limited by the extraction of sample DNA prior to genetic identification (27). Although LAMP technology exhibits some tolerance to impurities, nucleic acid extraction remains an essential step in LAMP detection, even when employing simple boiling or lysis solution lysis methods (28). Effective nucleic acid extraction can significantly enhance the accuracy and sensitivity of the detection, thereby expanding the application scope of NMED technology.

We have demonstrated that endonuclease cleavage of the labeling probe, followed by an immunological reaction, can initiate a second round of signal amplification for the LAMP amplicon. This method improves the detection sensitivity by over 10 times and allows for easier identification by the naked eye, aided by the use of colloidal gold chromatographic test paper. The detection sensitivity is close to the level of detecting single molecules, and the entire detection process only requires basic temperature control equipment. Compared to qPCR, the detection coincidence rate can reach 100%.

At present, the NMED method has been established in the laboratory and used in nearby veterinary hospitals and other locations for CPV detection, yielding positive outcomes. The method has several advantages, including simple sampling, high sensitivity, intuitive results, and no requirement for expensive equipment. The establishment of this method has commercial potential and offers a novel approach and concept for the future development of clinical detection of pathogenic microorganisms.

## ACKNOWLEDGMENTS

This work was supported by the central government-guided local science, Anyang Key Research and Development Project (grant no. 2023C01SF172) and Doctoral Research Start-up Fund Project (grant no. BSJ2021019).

S.W., L.Z., and Y.W. contributed to the study conception and design. S.M., Y.X., Z.L., Q.L., and M.F. performed material preparation and experiment operation. W.Z., Y.W., and S.L. performed data collection and analysis. S.W. wrote the main manuscript text. All authors commented on previous versions of the manuscript and read and approved the final manuscript.

## AUTHOR AFFILIATIONS

[1]Department of Biotechnology, Anyang Institute of Technology, Anyang, Henan, China
[2]College of Animal Science and Veterinary Medicine, Henan Institute of Science and Technology, Xinxiang, Henan, China
[3]Anyang Kindstar Global Medical Laboratory Ltd, Anyang, Henan, China

## AUTHOR ORCIDs

Shaoting Weng  http://orcid.org/0000-0001-8953-4436
Longfei Zhang  http://orcid.org/0009-0006-3062-6161
Yao Wang  http://orcid.org/0009-0009-4808-8017

## FUNDING

| Funder | Grant(s) | Author(s) |
|--------|----------|-----------|
| the central government guided local science | | Yao Wang |
| Anyang Institute of Technology (AYIT) | BSJ2021019 | Shaoting Weng |
| Anyang Science and Technology Information Network (Anyang Science and Technology Bureau) | 2023C01SF172 | Shaoting Weng |

## ADDITIONAL FILES

The following material is available online.

Open Peer Review

**PEER REVIEW HISTORY (review-history.pdf).** An accounting of the reviewer comments and feedback.

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
