## [Reviewer comments · Microbiology Spectrum]

Microbiology Spectrum

Towards establishing a rapid constant temperature detection method for canine parvovirus based on endonuclease activities

shaoting Weng, Shengming Ma, Yueteng Xing, Wenhui Zhang, Yinrong Wu, Mengyao Fu, Zhongyi Luo, Qiuying Li, Sen Lin, Longfei Zhang, and Yao Wang

Corresponding Author(s): Longfei Zhang, Henan Institute of Science and Technology

Review Timeline:

Submission Date:	December 16, 2023
Editorial Decision:	April 22, 2024
Revision Received:	May 12, 2024
Editorial Decision:	June 28, 2024
Revision Received:	July 20, 2024
Accepted:	August 13, 2024

Editor: Clinton Jones

Reviewer(s): The reviewers have opted to remain anonymous.

Transaction Report:

DOI: <https://doi.org/10.1128/spectrum.04222-23>

Re: Spectrum04222-23 (Towards establishing a rapid constant temperature detection method for canine parvovirus based on endonuclease activities)

Dear Dr. Longfei Zhang:

Thank you for the privilege of reviewing your work. Below you will find my comments, instructions from the Spectrum editorial office, and the reviewer comments.

Revision Guidelines

Sincerely,
Clinton Jones
Editor
Microbiology Spectrum

Reviewer #2 (Comments for the Author):

Canine parvovirus is a pathogen of veterinary significance. This manuscript describes the invention and prototype testing of a point-of-care diagnostic test combining LAMP, endonuclease probe cleavage, and lateral flow chromatography. With further development, validation, and standardization, it has potential as a product (at least from a scientific point of view).

I would have more enthusiasm for this manuscript with more stringent controls and sample sizes. As an example, one assay established a limit of detection as 10 copies, but detection happened in every lane and nothing below 10 copies was used. Maybe the limit of detection is lower, or maybe unrelated DNA is being amplified such that false-positives crop up after extensive amplification. For clinical validation, 10 suspected parvo cases were tested of which 8 were qPCR and LAMP/NMED positive and 2 were qPCR and LAMP/NMED negative. Having only 2 negatives does not give much confidence in the assay specificity, and we don't know what the rate of positives is in non-gastroenteritic controls. So, expanding the clinical validation would strengthen confidence in the assay.

Minor comments:

This manuscript claims to combine RT-LAMP with enzyme digestion, but there is no reverse transcription. So isn't this normal (DNA) LAMP rather than RT-LAMP?

I. 26: Rather than 100% accuracy, it would be more accurate to say 100% agreement with qPCR. The insensitivities of both these assays should be correlated.

I. 32: Wrong genus name; also please consult ICTV orthography guidelines to fix italicization

I. 33-34: This is incorrect; the virus once known as CPV-1 (also "minute virus of canines") is a bocavirus in a different genus and is NOT a form of canine parvovirus. The viruses aren't that closely related. All canine parvoviruses are a form of CPV-2. This confusing nomenclature dates to the era before we had good assays and sequencing.

I. 41: Canine parvovirus is detected by these assays, not canine parvovirus disease. The disease is clinically detected.

I. 43: What is "blood routine"?

I. 54: nuclear capsid -> capsid

II. 133, 143, &c.: What is the buffer composition?

Fig. 2: Which lane corresponds to which primer set is not described. It would have been nice to see control lanes with unrelated template such as dog gDNA or adeno-associated virus.

In general it would be easier to read if lanes were labeled directly rather than by keys when the label is short. For instance, in Fig. 3B, having "M 0 10 20 30 40 50 60" and writing that this means Marker and time in minutes would make the panel quicker to interpret.

Clinical characteristics of the tested dogs should be described.

Reviewer #3 (Comments for the Author):

The manuscript describes a new diagnostic method to detect canine parvovirus infection based on endonuclease activity. Indeed such applications are of potential interest and the results provided by the authors strongly suggest the feasibility of the proposed method. However, there are a number of uncertainties the authors should address before publication:

Although mentioned by the authors the benefit for the proposed approach as compared to qPCR remains largely unclear, since it requires multiple steps at different temperatures after isolation of the single-stranded virion DNA before the final read-out is available.

The authors chose different primers and probes in the VP2 region for the detection of CPV DNA with diverse outcome. A rationale to target the VP2 rather than the NS1 region remains vague and it is largely elusive, whether the approach is capable to detect all or a selection of the currently circulating CPV strains. In addition, the target region for qPCR as compared to the RT-LAMP is not mentioned.

To establish optimal conditions for RT-LAMP the authors used double-stranded plasmid DNA instead of single-stranded virion DNA derived from dog samples. It remains unclear whether this is of relevance or not.

Figure 6 (comparison of case samples) lacks annotations of T and C in the test strips. This is particularly disturbing regarding case G, which appears to be negative.

Response to Reviewers

- Upload point-by-point responses to the issues raised by the reviewers in a file named "Response to Reviewers," NOT in your cover letter.

Reply: I have addressed the reviewer's issues point-by-point and uploaded the responses.

- Upload a compare copy of the manuscript (without figures) as a "Marked-Up Manuscript" file.

Reply: I have uploaded a "Marked-Up Manuscript" file.

- Upload a clean .DOC/.DOCX version of the revised manuscript and remove the previous version.

Reply: I have uploaded a clean .DOC/.DOCX version of the revised manuscript

- Each figure must be uploaded as a separate, editable, high-resolution file (TIFF or EPS preferred), and any multipanel figures must be assembled into one file.

Reply: Each figure was saved individually in high resolution, editable, TIFF format. All figures were uploaded in one file.

- Any intended for posting by ASM should be uploaded with their legends separate from the main manuscript. You can combine all supplemental material into one file (preferred) or split it into a maximum of 10 files with all associated legends included.

Reply: I have sorted out the manuscript and the legends separately and uploaded them.

Reviewer #2 (Comments for the Author):

1. I would have more enthusiasm for this manuscript with more stringent controls and sample sizes. As an example, one assay established a limit of detection as 10 copies, but detection happened in every lane and nothing below 10 copies was used. Maybe the limit of detection is lower, or maybe unrelated DNA is being amplified such that false-positives crop up after extensive amplification.

Reply: Thank you for your comments. The main reason why you have this question is that there is a typo in Figure 5. From the article's context, it is evident that the smallest copy of dilution in our experiment is 1 (10^0). The other authors mistakenly took 10^0 as

0 when drafting the manuscript, which is a typo and we have corrected it. As can be seen from Figure 5D, the detection limit is 1 copy.

2. For clinical validation, 10 suspected parvo cases were tested of which 8 were qPCR and LAMP/NMED positive and 2 were qPCR and LAMP/NMED negative. Having only 2 negatives does not give much confidence in the assay specificity, and we don't know what the rate of positives is in non-gastroenteritic controls. So, expanding the clinical validation would strengthen confidence in the assay.

Reply: Thank you for your suggestion. We have increased the number of samples in response to your comments. This includes testing healthy samples, non-gastroenteritis samples, canine distemper virus samples, and parainfluenza virus samples. The supplementary results are shown in Figure 6. The experimental results are satisfactory, which proves that the nucleic acid detection technology is suitable for CPV clinical detection.

Minor comments:

3. This manuscript claims to combine RT-LAMP with enzyme digestion, but there is no reverse transcription. So isn't this normal (DNA) LAMP rather than RT-LAMP?

Reply: I'm Sorry, Parvovirus is a DNA virus and does not require reverse transcription. This is a typo on our part, where RT-LAMP is understood as room temperature LAMP. We have removed the RT- from the RT-LAMP in the manuscript.

4. 26: Rather than 100% accuracy, it would be more accurate to say 100% agreement with qPCR. The insensitivities of both these assays should be correlated.

Reply: Thank you for your suggestion, it has been modified.

5. 32: Wrong genus name; also please consult ICTV orthography guidelines to fix it alicization

Reply: Thank you for your careful guidance, it has been modified.

6. 33-34: This is incorrect; the virus once known as CPV-1 (also "minute virus of canines") is a bocavirus in a different genus and is NOT a form of canine parvovirus. The viruses aren't that closely related. All canine parvoviruses are a form of CPV-2. This confusing nomenclature dates to the era before we had good assays and sequencing.

Reply: Thanks for your scientific explanation, we have revised this sentence.

7. 41: Canine parvovirus is detected by these assays, not canine parvovirus disease. The disease is clinically detected.

Reply: Thank you for your careful guidance, it has been modified.

8. 43: What is "blood routine"?

Reply: I'm sorry I didn't make it clear, it's routine blood tests.

9. 54: nuclear capsid -> capsid

Reply: Thank you for your careful guidance, it has been modified.

10. 133, 143, &c.: What is the buffer composition?

Reply: Thank you for your suggestion, the buffers information have been marked.

11. Fig. 2: Which lane corresponds to which primer set is not described. It would have been nice to see control lanes with unrelated template such as dog gDNA or adeno-associated virus.

Reply: Thanks for your suggestion, we have detailed the primers for each lane in the notes in Figure 2. For the detection of unrelated templates, these were in clinical detection experiments. Since the figure was designed to test the feasibility of the method establishment, PCR products of CPV with definite quantity and stable reaction were used.

12. In general it would be easier to read if lanes were labeled directly rather than by keys when the label is short. For instance, in Fig. 3B, having "M 0 10 20 30 40 50 60" and writing that this means Marker and time in minutes would make the panel quicker

to interpret.

Reply: Thank you for your suggestion, but I found through the references that some articles are consistent with my labeling method. To ensure consistency with the other figure labels in this article, I want to keep this label.

13. Clinical characteristics of the tested dogs should be described.

Reply: Thank you for your suggestion, because research focuses on the establishment of nucleic acid detection methods for CPV through molecular biological techniques. The samples we collected were obtained by the veterinarians at the pet hospital, and there was no detailed one-to-one correspondence between the samples and the clinical symptoms of the sick dogs. In the future, we will pay attention to collecting complete sample information.

Reviewer #3 (Comments for the Author):

The manuscript describes a new diagnostic method to detect canine parvovirus infection based on endonuclease activity. Indeed such applications are of potential interest and the results provided by the authors strongly suggest the feasibility of the proposed method. However, there are a number of uncertainties the authors should address before publication:

1. Although mentioned by the authors the benefit for the proposed approach as compared to qPCR remains largely unclear, since it requires multiple steps at different temperatures after isolation of the single-stranded virion DNA before the final read-out is available.

Reply: Thank you for your comments. The purpose of this study method, as compared to qPCR, is not to demonstrate that the method is superior to qPCR. But because the qPCR method is the gold standard for detecting viral nucleic acids. This study presents a new nucleic acid assay that compares qPCR exclusively to demonstrate its feasibility for detecting CPV based on sensitivity and specificity. Of course, compared with the qPCR method, it has the advantage of low equipment requirements.

2. The authors chose different primers and probes in the VP2 region for the detection of CPV DNA with diverse outcome. A rationale to target the VP2 rather than the NS1 region remains vague and it is largely elusive, whether the approach is capable to detect all or a selection of the currently circulating CPV strains. In addition, the target region for qPCR as compared to the RT-LAMP is not mentioned.

Reply: Thank you for your comments. One of the reasons for choosing VP2 is that it is the coat protein gene of CPV, which is highly conserved and important and plays a crucial role in the antigenicity and virulence of the virus. Secondly, most of the articles on nucleic acid detection of CPV are designed with VP2 primers. The designed region for qPCR is 469 bp downstream of the 5' end of VP2, and the primer sequences are shown in Table 1.

3. To establish optimal conditions for RT-LAMP the authors used double-stranded plasmid DNA instead of single-stranded virion DNA derived from dog samples. It remains unclear whether this is of relevance or not.

Reply: Thank you for your careful consideration. I think they are pertinent from the results of method establishment to clinical identification in this study. In this study, it is necessary to use plasmid DNA to screen the optimal LAMP conditions, because the amount of DNA is known and its stability is high in the experiment. Plasmid DNA is commonly utilized in various articles to determine the detection limits of tests, including the recent novel coronavirus test.

4. Figure 6 (comparison of case samples) lacks annotations of T and C in the test strips. This is particularly disturbing regarding case G, which appears to be negative.

Answer: Thank you for your comments. The labels for T and C have been added in Figure 6 and its notes. Case G was the positive result, where the red position indicated the T-line, and a very faint red line below the T-line indicated the C-line. As shown in the following figure.

Re: Spectrum04222-23R1 (Towards establishing a rapid constant temperature detection method for canine parvovirus based on endonuclease activities)

Dear Dr. Longfei Zhang:

Thank you for the privilege of reviewing your work. Below you will find my comments, instructions from the Spectrum editorial office, and the reviewer comments.

Please return the manuscript within 60 days; if you cannot complete the modification within this time period, please contact me. It is essential to address all of the concerns raised by Reviewer #2. If you do not wish to modify the manuscript and prefer to submit it to another journal, notify me immediately so that the manuscript may be formally withdrawn from consideration by Spectrum.

Revision Guidelines

Sincerely,
Clinton Jones
Editor
Microbiology Spectrum

Reviewer #2 (Comments for the Author):

The manuscript was much improved with additional experimentation.

I. 27 "samples were detected" -> maybe "samples were evaluated"; "detected" can imply virus presence

Reviewer #4 (Comments for the Author):

Towards establishing a rapid constant temperature detection method for canine parvovirus based on endonuclease activities present a new method with high sensitivity for the detection of CP, the authors tested the system on 20 real samples, were 15 results to be positive for the virus. The authors indicate that the sensitivity of the new method is comparable to qPCR, and that it has the potential to be package for veterinary use. I must say that I do not recall to have participated on the first revision of this article, and I am quite surprised to find that correctly designed and logical experiments set of experiments with validation in real samples, is so badly presented. The figures lack resolution, the migration of the DNA markers is not labeled, figures with several panels were not assembled as 1 figure, and files for each panel are presented. Below I am listing all my observations of this manuscript.

Introduction:

Line 57 LAMP - define the acronym

Lines 59 RPA :-define the acronym

Lines 645 DETECTR -define)

Lines 47-76 Is confusing that the authors were first talking about CPV detection methods, and then move to other methods that were not specifically designed for this virus, then they should clarify that these are novel methods for viral detection, in general. Maybe they could separate this to another paragraph.

The authors should finish the introduction with the final findings about accuracy, sensibility and comparability against current methods.

MATERIALS AND METHODS

Lines 96 were should be was

Lines 188 DNA extraction: This is a limiting step, if DNA extraction is not done properly, then the LAMP reaction could fail even from positive samples, the authors did not discuss this limitation of the system, in the discussion section. Just a question, Do they plan to test if their system works with direct samples?

Line 193 10 minutes, should be 10 min.

RESULTS

The figures are not in one sheet, and there is one file for every panel of every figure, this is really complex to evaluate. The author must compile very figures in one sheet, with appropriate resolution. Also, figures are badly are really badly presented. Fig 2A and B do not have the label of the ladder indicating the size of the DNA, none of the figures does. In figure 2B there is not clear differences between the lines. The figure legend is quite difficult to understand, the author must made better figures labeling them correctly, so the reader can understand what is in each lines without having to read the legend.

Line 208 "CVP-2 was significantly better than that of the CPV3 (Fig. 2C)". What do the authors mean by significantly better? Did they quantify the intensity of the bands? Or do they mean that the difference is clear?

Lines 208-209 "It was indicated that the enzyme-cut products were identified." What is the meaning of this phrase?

Figure 3A Once again labeling just with numbers is quite confusing, especially if the reader has only the gel picture and the text, I suggest replace the numbers above the gel with the temperatures. Also it is very strange that results are described from the lower temperature to the higher one, but the gel was loaded from the higher temperature to take lower one.

Figure 3B the ladder must be labeled, and the numbers should be replaced by the times were samples were collected.

Line 232 "Cruiser exhibited a more effective cleavage compared to T7E I (Fig. 4A)". This is not clear from the figure, since is very pixelated. How did the authors determine this? Did they quantify the amount of cleavage product? What is the difference between lines 5 and 6? That is not stated in the legend, and again number should be replaced with conditions.

Figure 4B: T and C are not labeled in the figure.

Line 237 AnstivitySensitivity?

Figures 5A and B : As all the figures, all partes should be presented in one file. Dilutions in the figures are indicated with dilution number, and if in the text the authors are indicating the copy numbers, then that should be the labeling that they must use in the figure, so the reader does not get lost.

Am I correct to assume that dilution 8 is 1 copy of the standard plasmid?

The accuracy and reliability of NMED assay and Figure 6: It is quite disturbing having to open 20 different files to understand this section. The authors must do an effort to make figures amicable to the reader. Did they really perform 20 separated qPCR assays rather than run several samples together and that is why they are presenting this in such a manner.

Response to Reviewers

Reviewer #2 (Comments for the Author):

1. 27 "samples were detected" -> maybe "samples were evaluated"; "detected" can imply virus presence

Reply: Thanks for your valuable suggestion. I have incorporated your suggestions into the revisions.

Reviewer #4 (Comments for the Author):

INTRODUCTION:

1. Line 57 LAMP - define the acronym

Reply: The acronym LAMP has been defined.

2. Lines 59 RPA :-define the acronym

Reply: The acronym RPA has been defined.

3. Lines 645 DETECTR -define)

Reply: The acronym DETECTR has been defined.

4. Lines 47-76 Is confusing that the authors were first talking about CPV detection methods, and then move to other methods that were not specifically designed for this virus, then they should clarify that these are novel methods for viral detection, in general. Maybe they could separate this to another paragraph.

Reply: Thank you for your careful corrections. The introduction of the CPV detection method was actually intended to illustrate the application of isothermal detection methods, which I did not make clear. Therefore, I have added a general statement at the beginning of this section. As there are no reports on the application of CRISPR-Dx in CPV, I can only cite typical articles. I have separated this section from the previous one as suggested.

5. The authors should finish the introduction with the final findings about accuracy, sensibility and comparability against current methods.

Reply: Thank you for your suggestion. I have included the relevant content at the end of the introduction.

MATERIALS AND METHODS

6. Lines 96 were should be was

Reply: I should have used "were". The reagents include The 2×Taq Master Mix and ChamQ Universal SYBR qPCR master Mix.

7. Lines 188 DNA extraction: This is a limiting step, if DNA extraction is not done properly, then the LAMP reaction could fail even from positive samples, the authors did not discuss this limitation of the system, in the discussion section. Just a question, Do they plan to test if their system works with direct samples?

Reply: That's a great question. To validate the applicability of this system to direct samples, we conducted several experiments, which showed that direct samples are not suitable. DNA extraction is a crucial step. Besides the Kit extraction method mentioned in the article, there are two simpler extraction methods: 1. Viral lysis buffer method: Treat viruses with viral lysis buffer, then centrifuge at high speed, and collect

the supernatant. This method is relatively common, and there are many commercial products available on the market. 2. Boiling water boiling method: Boil the virus sample in boiling water for 10 minutes, then centrifuge at high speed, and collect the supernatant. Regardless of the method used, virus lysis is required for detection using this system. We have added these details to the discussion section.

8. Line 193 10 minutes, should be 10 min.

Reply:I have made the modifications.

RESULTS

9. The figures are not in one sheet, and there is one file for every panel of every figure, this is really complex to evaluate. The author must compile very figures in one sheet, with appropriate resolution. Also, figures are badly are really badly presented.

Reply:I have integrated the figures and ensured that each figure has 300 pixels per inch.

10. Fig 2A and B do not have the label of the ladder indicating the size of the DNA, none of the figures does. In figure 2B there is not clear differences between the lines. The figure legend is quite difficult to understand, the author must made better figures labeling them correctly, so the reader can understand what is in each lines without having to read the legend.

Reply:Thank you for your valuable feedback. I have added captions to the figures in the article. Regarding the restriction enzyme results in Figure 2B, the differences compared to the control group are quite noticeable. We may have been lacking in our labeling explanations, leading to a lack of full understanding of the figure results. I have now made corrections to hopefully present the experimental results more clearly.

11. Line 208 "CVP-2 was significantly better than that of the CPV3 (Fig. 2C)". What do the authors mean by significantly better? Did they quantify the intensity of the bands? Or do they mean that the difference is clear?

Reply:Thank you for your valuable feedback. We understand the issue you raised and agree that the term "significantly" was not used appropriately in the article, and simply reflects a difference. The writer of this paper have reversed the order of CPV-2 and CPV-3. It was that CPV-3 actually had higher value than CPV-2. The following content indicates that the experiment was conducted using the more efficient CPV-3. Regarding the quantification of band intensity, our paper utilizes Photoshop for data analysis. Previously, due to the simplicity of our analytical method, the referenced published articles did not provide detailed descriptions of their analysis methods. Therefore, we did not elaborate on this aspect in our article. We have adopted your suggestion and made the corresponding modifications to the paper.

12. Lines 208-209 "It was indicated that the enzyme-cut products were identified." What is the meaning of this phrase?

Reply:I apologize for the lack of clarity in my previous explanation. The statement you mentioned is based on the experimental results from Figure 2C. The results of the colloidal gold experiment demonstrate that the double-labeled probes were recognized by the cleaved by lateral chromatographic test paper after cleavage by T7E I enzyme. I have revised the relevant content to ensure a clear explanation.

13. Figure 3A Once again labeling just with numbers is quite confusing, especially if the reader has only the gel picture and the text, I suggest replace the numbers above the gel with the temperatures. Also it is very strange that results are described from the lower temperature to the higher one, but the gel was loaded from the higher temperature to take lower one.

Reply:Following your suggestions, I have revised the content of the figure. As the original author and the experiment executor were different individuals, their logical order differed. I have made modifications to ensure consistency in the logical sequence.

14. Figure 3B the ladder must be labeled, and the numbers should be replaced by the times were samples were collected.

Reply:I have revised the content in the figure based on your suggestions.

15. Line 232 "Cruiser exhibited a more effective cleavage compared to T7E I (Fig. 4A)". This is not clear from the figure, since is very pixelated. How did the authors determine this? Did they quantify the amount of cleavage product? What is the difference between lines 5 and 6? That is not stated in the legend, and again number should be replaced with conditions.

Reply:Thank you for your valuable suggestions. For Figure 4A, we did not perform a gray-scale analysis because the LAMP products were complex. Although the gel image clearly showed the effect of cleavage, it was difficult to fully analyze the amount of change after cleavage. However, the figure clearly showed that Cruiser had a stronger cleavage effect than T7E I. The difference is clear. Lines 5 and 6 were the results of two separate Cruiser enzyme cleaved, which were illustrated in the figure note. The number have been replaced with conditions.

16. Figure 4B: T and C are not labeled in the figure.

Reply:I've labeled it.

17. Line 237 AnstivitySensitivity?

Reply:Thanks for your careful correction, I have modified it.

18. Figures 5A and B : As all the figures, all partes should be presented in one file. Dilutions in the figures are indicated with dilution number, and if in the text the authors are indicating the copy numbers, then that should be the labeling that they must use in the figure, so the reader does not get lost.

Am I correct to assume that dilution 8 is 1 copy of the standard plasmid?

Reply:Thank you for your suggestion. I have revised the content in the figure.The statement you pointed out about the 8-fold dilution representing 1 copy of the standard plasmid is inaccurate. There are standard methods for calculating plasmid size and viral copy number. The formula for converting plasmid concentration to copy number is as follows: Copy number = (Plasmid concentration (ng/μL) ÷ Plasmid length (bp)) × 6.022 × 10²³. Include: **Plasmid Concentration (ng/μL)** is determined by measuring the absorbance of the DNA solution at 260 nm (A₂₆₀). The specific formula used is: Plasmid Concentration (ng/μL) = A₂₆₀ Absorbance Value × Dilution Factor of the Sample × 50 (where 50 is the proportionality constant for a DNA concentration of 1 μg/mL). **Plasmid Length (bp)** refers to the total number of base

pairs on the plasmid, expressed in base pairs (bp). 6.022×10^{23} represents Avogadro's constant, which denotes the number of basic units contained in one mole of a substance. We calculated the highest copy number using an algorithm and performed 10-fold serial dilutions until 1 copy was reached. Hopefully, this revision and explanation clarify the experiment for you.

19. The accuracy and reliability of NMED assay and Figure 6: It is quite disturbing having to open 20 different files to understand this section. The authors must do an effort to make figures amicable to the reader. Did they really perform 20 separated qPCR assays rather than run several samples together and that is why they are presenting this in such a manner.

Reply: Thank you for your suggestion. We have integrated the results of 20 experiments from Figure 6 into a figure. As the sample collection spanned a period of time, we performed 9 qPCR analyses (including replicates), each analysis involving approximately 6-10 samples. The figures are presented in this manner primarily to ensure consistency and comparability, referencing the methods used in relevant published articles. Furthermore, the comparative results from both qPCR and NMED methods allow for a more clear and direct presentation.

Re: Spectrum04222-23R2 (Towards establishing a rapid constant temperature detection method for canine parvovirus based on endonuclease activities)

Dear Dr. Longfei Zhang:

Your manuscript has been accepted, and I am forwarding it to the ASM production staff for publication. Your paper will first be checked to make sure all elements meet the technical requirements. ASM staff will contact you if anything needs to be revised before copyediting and production can begin. Otherwise, you will be notified when your proofs are ready to be viewed.

Sincerely,
Clinton Jones
Editor
Microbiology Spectrum

Reviewer #4 (Comments for the Author):

The authors have answered all my concerns, the figures are now clear and properly labeled, the text reads much better, thank you for taking my suggestions into consideration.